# Learn more, but bother less: parameter efficient continual learning

**Fuli Qiao**
Pennsylvania State University
fvq5015@psu.edu

**Mehrdad Mahdavi**
Pennsylvania State University
mzm616@psu.edu

## Abstract

Large Language Models (LLMs) have demonstrated profound capabilities due to their extensive pre-training on diverse corpora. However, LLMs often struggle with catastrophic forgetting when engaged in sequential task learning. In this paper, we propose a novel parameter-efficient approach for continual learning in LLMs, which empirically investigates knowledge transfer from previously learned tasks to new tasks through low-rank matrix parameters, enhancing the learning of new tasks without significant interference. Our method employs sensitivity-based analysis of low-rank matrix parameters to identify knowledge-specific parameters between sequential tasks, which are used to initialize the low-rank matrix parameters in new tasks. To maintain orthogonality and minimize forgetting, we further involve the gradient projection technique that keeps the low-rank subspaces of each new task orthogonal to those of previous tasks. Our experimental results on continual learning benchmarks validate the efficacy of our proposed method, which outperforms existing state-of-the-art methods in reducing forgetting, enhancing task performance, and preserving the model's ability to generalize to unseen tasks.

## 1 Introduction

Large Language Models (LLMs) [1, 5, 32, 40] have demonstrated exceptional performance across a broad spectrum of tasks, significantly revolutionizing the landscape in diverse areas driven by artificial intelligence. Full fine-tuning all the parameters of the pre-trained models becomes prohibitive in adapting pre-trained models to a large number of downstream tasks. Thus, following LoRA [11], multiple variants [24, 50] of low-rank adaptation have been proposed to prompt parameter-efficient learning for LLMs. AdaLoRA [50] enhances the flexibility of low-rank matrices for various device budgets by parameterizing the increments through singular value decomposition. It generalizes the matrices, enabling layer-specific rank adjustments to meet different parameter constraints.

Despite the tremendous success of pre-trained models on static tasks, learning multiple sequential tasks, known as continual learning (CL), still poses significant challenges [45]. There are two primary challenges: (i) overcoming catastrophic forgetting, where a model's performance on previous tasks significantly deteriorates upon training with new data [28, 34], and (ii) facilitating forward transfer, where knowledge from old tasks is harnessed to enhance the learning of new tasks. In the context of LLMs, continual learning extends beyond merely enhancing linguistic and reasoning capabilities and it encompasses a multi-faceted process, including continual pertaining [12], continual instruction [52], and continual alignment [49].

While existing parameter-efficient tuning (PET) methods for CL [27, 31, 39, 43] have primarily focused on mitigating the forgetting issue, they often overlook the equally important objective of facilitating forward knowledge transfer. This transfer is crucial as it allows the model to leverage previously acquired knowledge to better generalize to new tasks or domains. For instance, O-LoRA [43], operating within the PET framework, proposes an orthogonal low-rank adaptation

38th Conference on Neural Information Processing Systems (NeurIPS 2024).

for continual instruction. This method incrementally learns new tasks in an orthogonal subspace, ensuring that the LoRA parameters from previous tasks remain fixed to minimize forgetting. Besides, compared to full fine-tuning, LoRA inherently forgets less of the source domain, serving as a form of regularization [2]. However, O-LoRA does not explicitly address knowledge transfer across different tasks.

Focusing on knowledge transfer among tasks, many existing non-PET knowledge transfer methods in CL, such as Progressive Network [37] which tackles forward knowledge transfer, and CUBER [22] which employs backward knowledge transfer, while having distinctive approaches to managing knowledge across tasks, they are not directly applicable to the continual learning setting in the PET framework due to their prohibitive computational costs. Although there have been recent attempts at parameter-efficient fine-tuning of LLMs, these methods have failed when applied to CL. For instance, while the parametric knowledge transfer paradigm [53] in PET harnesses the rich knowledge embedded within a teacher's parameters by extracting task-specific parameters and injecting them into a student model via sensitivity metrics, such methods do not exist in CL for LLMs.

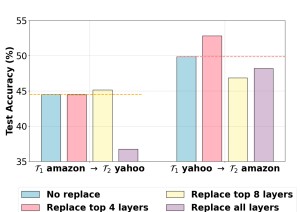

Figure 1: Comparison performance of the model after training task $\mathcal{T}_2$ with different layers replacement.

To investigate the impact of leveraging parametric knowledge in the incremental learning of LoRA parameters on a new task, we conduct an experiment where layers from the previous task's LoRA-based layers are simply replaced at the training initialization of the new task's LoRA layers, without a predefined strategy for selecting key parameters. The results, presented in Tab. 1 and Fig. 1 using the pre-trained model T5-large [32] with incremental LoRAs for Amazon Reviews and Yahoo Answers [51], indicate that this simple layer replacement can enhance performance. Notably, Tab. 1 reveals that the best replacement strategy significantly improves the performance of both the new task $\mathcal{T}_2$ and the old task $\mathcal{T}_1$ compared to scenarios with no replacement.

Table 1: Testing accuracy of $\mathcal{T}_1$ and $\mathcal{T}_2$ after training $\mathcal{T}_2$ with different layer replacements, highlighting the best-performing strategy as shown in Fig. 1.

| $\mathcal{T}_1$ (amazon) $\rightarrow \mathcal{T}_2$ (yahoo) | | | |
|---|---|---|---|
| no | top 4 | **top 9** | all |
| $\mathcal{T}_1$ | 15.4 | 15.0 | **16.6** | 0.2 |
| $\mathcal{T}_2$ | 73.6 | 73.9 | **73.7** | 73.3 |
| $\mathcal{T}_1$ (yahoo) $\rightarrow \mathcal{T}_2$ (amazon) | | | |
| no | **top 4** | top 9 | all |
| $\mathcal{T}_1$ | 46.8 | **50.7** | 39.1 | 42.5 |
| $\mathcal{T}_2$ | 52.9 | **54.9** | 54.5 | 53.9 |

The aforementioned observations motivate that an ideal approach for continual learning of LLMs in the PET framework should take the best of both worlds by simultaneously overcoming catastrophic forgetting and promoting forward transfer for enhanced generalization across a continual stream of tasks. Motivated by these needs, we seek to explore a new dimension in CL for LLMs:

*How can we effectively inject knowledge from previous tasks into new tasks (for improving* ***generalization****) while maintaining the orthogonality of each task's low-rank subspaces (for mitigating* ***forgetting****) to facilitate parameter-efficient continual learning?*

To answer this question, we propose a novel *all-for-all parameter-efficient* approach for continual learning in LLMs, which empirically investigates knowledge transfer from previously learned tasks to new tasks using low-rank matrix parameters evaluated by sensitivity scores while maintaining orthogonality via gradient projection techniques. We name this method LB-CL (**L**earn more but **b**other less **C**ontinual **L**earning). Specifically, LB-CL first calculates sensitivity metrics for SVD-based low-rank parameters of previous tasks to guide the injection of parametric knowledge into new task parameters. Then, it supports incremental learning of new tasks in orthogonal subspaces by preserving low-rank parameters of previous tasks. When transferring knowledge, LB-CL prioritizes low-rank triplets (consisting of a singular value and its corresponding singular vectors) from past tasks based on their sensitivity scores, enabling the new task to learn more from higher-scoring triplets. To preserve the orthogonality of low-rank subspaces, we project the gradients of the new task onto the subspaces formed by previously learned low-rank triplets, encouraging the new task to deviate appropriately from more impactful triplets. Our experimental results validate that LB-CL surpasses previous state-of-the-art methods on standard continual learning benchmarks. Moreover, our analysis highlights the critical role of initialization strategies in promoting generalization through effective parametric knowledge transfer, while also using low-rank orthogonal gradient projection to mitigate catastrophic forgetting.

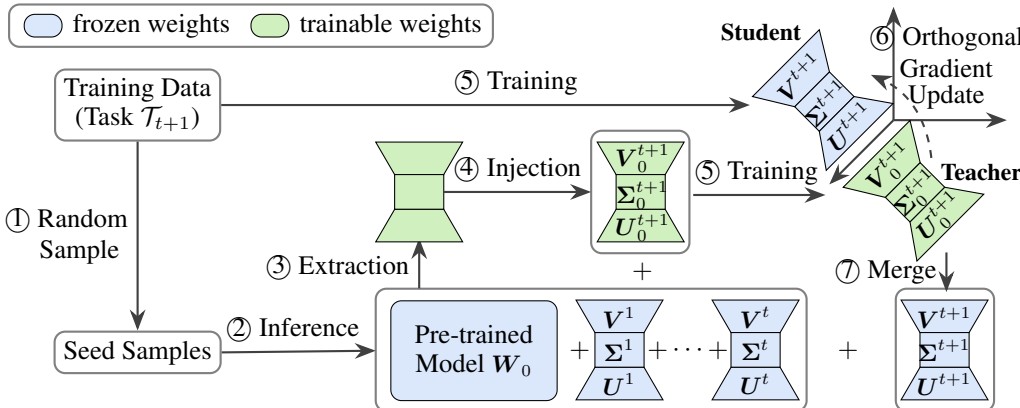

Figure 2: Overview of our LB-CL framework. Starting with the pre-trained model including SVD weights of previous tasks, sensitivity metrics are calculated using a set of seed samples, facilitating the extraction of task-specific knowledge. Subsequently, the extracted layer triplets initialize SVD weights for the new task. Then, the new task is trained in an orthogonal subspace, employing orthogonal gradient projection to minimize forgetting.

■ **Summary of Contributions.** This paper makes three key contributions: (1) A novel parameter-efficient continual learning framework for LLMs that balances generalization through parametric knowledge transfer and the mitigation of forgetting through low-rank orthogonal subspace learning for new tasks; (2) Through comprehensive evaluations, our method demonstrates superior performance over existing state-of-the-art approaches on standard continual learning benchmarks; and (3) We provide an in-depth analysis that deepens our understanding of the dynamics of parametric knowledge transfer within continual learning for LLMs, pinpointing critical factors that drive its effectiveness.

## 2 Continual Learning Maestro: Learn More but Bother Less

We consider a continual learning scenario where the learner is presented with a sequence of tasks $\{\mathcal{T}_1, \mathcal{T}_2, \ldots, \mathcal{T}_T\}$ over time. Each task $\mathcal{T}_k$ is associated with a data distribution $\mathcal{D}_k$ and contains a separate target dataset $\mathcal{S}_k = \{(\boldsymbol{x}_{k,i}, y_{k,i})\}_{i=1}^{n_t}$ where $\boldsymbol{x}_{k,i} \in \mathcal{X}_k$ and $y_{k,i} \in \mathcal{Y}_k$. The goal of continual learning is to find a set of parameters $\boldsymbol{\theta} \in \Theta$ that can effectively solve all tasks up to the current task $\mathcal{T}_k$, while minimizing catastrophic forgetting of previously learned tasks. In continual learning of LLMs, we are given a pre-trained model $\boldsymbol{W}_0$, and would like to continually fine-tune on a sequence of tasks, utilizing the incremental SVD-based low-rank matrix parameters $\boldsymbol{U}^k \boldsymbol{\Sigma}^k \boldsymbol{V}^k$ to fine-tune on task $\mathcal{T}_k$, where $\boldsymbol{U}^k \in \mathbb{R}^{d_1 \times r}$ and $\boldsymbol{V}^k \in \mathbb{R}^{r \times d_2}$ represent the left/right singular vectors and the diagonal matrix $\boldsymbol{\Sigma}^k \in \mathbb{R}^{r \times r}$ contains the singular values $\{\lambda_i\}_{1 \leq i \leq r}$ with $r \ll \min(d_1, d_2)$. To enforce the orthogonality of $\boldsymbol{U}$ and $\boldsymbol{V}$, i.e. $\boldsymbol{U}^\top \boldsymbol{U} = \boldsymbol{V} \boldsymbol{V}^\top = \boldsymbol{I}$, we use the regularizer: $\mathcal{R}(\boldsymbol{U}, \boldsymbol{V}) = \|\boldsymbol{U}^\top \boldsymbol{U} - \boldsymbol{I}\|_{\mathrm{F}}^2 + \|\boldsymbol{V} \boldsymbol{V}^\top - \boldsymbol{I}\|_{\mathrm{F}}^2$. The continual learning model parameters after fine-tuning on task $\mathcal{T}_k$ is: $\boldsymbol{\theta}_k = \boldsymbol{W}_0 + \sum_{m=1}^{k} \boldsymbol{U}^m \boldsymbol{\Sigma}^m \boldsymbol{V}^m$. Our continual learning goal is to optimize the following objective across all tasks:

$$\max_{\boldsymbol{\theta}} \sum_{k=1}^{T} \sum_{(\boldsymbol{x}, y) \in \mathcal{T}_k} \log p_{\boldsymbol{\theta}}(y | \boldsymbol{x}), \tag{1}$$

where $\boldsymbol{\theta} = \boldsymbol{W}_0 + \sum_{k=1}^{T} \boldsymbol{U}^k \boldsymbol{\Sigma}^k \boldsymbol{V}^k$. Our method contains two important stages: (i) Learning from knowledge extraction and injection, which transfers knowledge from previously learned tasks to new tasks by incremental SVD triplet sensitivity metric; (ii) Training in Orthogonal Subspaces, which keeps the low-rank subspaces of new tasks orthogonal to those of old tasks. The detailed description of our proposed method, LB-CL, is shown in Fig. 2.

■ **Generalization and Forgetting Tradeoff of Low-rank Finetuning.** Before delving into the detailed steps of proposed algorithm, here we motivate these steps by investigating the forgetting and generalization tradeoff of SVD based low-rank parameter-efficient methods in CL. The performance of a CL algorithm is typically evaluated using two key metrics [3, 4, 16, 21, 23, 25] as defined below: (i) *Forgetting error*: which quantifies the performance degradation on previously learned tasks after acquiring a new task formulated as:

$$\mathcal{F}(\boldsymbol{\theta}_1,\ldots,\boldsymbol{\theta}_T) = \sum_{t=1}^{T-1} \mathcal{L}_t(\boldsymbol{\theta}_T) - \mathcal{L}_t(\boldsymbol{\theta}_t) \tag{2}$$

where $\boldsymbol{\theta}_t = \boldsymbol{W}_0 + \sum_{k=1}^{t} \boldsymbol{U}^k \boldsymbol{\Sigma}^k \boldsymbol{V}^k$, $\mathcal{L}_t(\cdot)$ is the generalization error on task $\mathcal{T}_t$, and $\mathcal{L}_t(\boldsymbol{\theta}_T) - \mathcal{L}_t(\boldsymbol{\theta}_t)$ is the performance degradation (forgetting) on tasks $\mathcal{T}_t$ between the model after training on task $\mathcal{T}_t$ and the model after training on the final task $\mathcal{T}_T$.

(ii) *Generalization error*: which measures the algorithm's capability to effectively learn a new task while preserving the knowledge gained from prior tasks defined as:

$$\mathcal{I}(\boldsymbol{\theta}_1,\ldots,\boldsymbol{\theta}_T) = \sum_{t=1}^{T} \mathcal{L}_t(\boldsymbol{\theta}_t) - \mathcal{L}_t(\boldsymbol{\theta}_t^*) \tag{3}$$

where $\mathcal{L}_t(\boldsymbol{\theta}_t) - \mathcal{L}_t(\boldsymbol{\theta}_t^*)$ measures the generalization gap between the CL model $\boldsymbol{\theta}_t$ and the optimally fine-tuned model $\boldsymbol{\theta}_t^* = \boldsymbol{W}_0 + \boldsymbol{U}_*^t \boldsymbol{\Sigma}_*^t \boldsymbol{V}_*^t$ on task $\mathcal{T}_t$. The generalization of final model on all tasks can be decomposed in terms of forgetting generalization errors as follows:

$$\sum_{t=1}^{T} \mathcal{L}_t(\boldsymbol{\theta}_T) - \mathcal{L}_t(\boldsymbol{\theta}_t^*) = \mathcal{F}(\boldsymbol{\theta}_1,\ldots,\boldsymbol{\theta}_T) + \mathcal{I}(\boldsymbol{\theta}_1,\ldots,\boldsymbol{\theta}_T) \tag{4}$$

It is crucial to note that the generalization error is based on the final model over all tasks, which may not fully capture each task-specific optimal setting since each task could potentially achieve better performance metrics when fine-tuned individually. Given the complexities in addressing our generalization error due to limited theoretical support, we utilize initialization parameters to further decompose the per-task generalization term on the left-hand side of Eq. 4. This involves measuring the performance difference between the CL model $\boldsymbol{\theta}_t$ after learning on tasks $\mathcal{T}_t$ and its initialization $\boldsymbol{\theta}_t^{(0)}$, evaluated on task $\mathcal{T}_t$ as:

$$\mathcal{L}_t(\boldsymbol{\theta}_T) - \mathcal{L}_t(\boldsymbol{\theta}_t^*) = \underbrace{\left( \mathcal{L}_t(\boldsymbol{\theta}_T) - \mathcal{L}_t(\boldsymbol{\theta}_t) \right)}_{\text{(I): forgetting error}} + \underbrace{\left( \mathcal{L}_t(\boldsymbol{\theta}_t) - \mathcal{L}_t(\boldsymbol{\theta}_t^{(0)}) \right)}_{\text{(II): improvement by fine-tuning}} + \underbrace{\left( \mathcal{L}_t(\boldsymbol{\theta}_t^{(0)}) - \mathcal{L}_t(\boldsymbol{\theta}_t^*) \right)}_{\text{(III): generalization of initial model}}$$

From this decomposition, it is clear that all three terms must be carefully calibrated to enhance the overall performance of CL algorithm. In the context of SVD-based low-rank fine-tuning, we have:

- The term (I) captures the forgetting error. It is evident that models that undergo minimal changes when fine-tuned to a new target domain will exhibit less forgetting of the source domain. This automatically holds for fine-tuning with low-rank updating methods as it acts as a regularization. This can be further amplified by fine-tuning in orthogonal subspace.

- The term (II) captures the performance of fine-tuning algorithm itself. Since fine-tuning with low-rank updates is not capable of approximating full fine-tuning accuracy [2], and in order to balance between stability (retaining old knowledge) and plasticity (acquiring new knowledge) we need to leverage methods such as orthogonal gradient projection to better align with current task.

- The term (III) captures the generalization of the initial model. Since the knowledge transfer from previous tasks happens through the initial model, naive averaging of low-rank models from previous tasks might entail a dramatically poor generalization on the current task, in particular when there is a larger domain shift among tasks. Consequently, the contribution of each task should be proportional to its similarity to the current task, measured by leveraging effective discrepancy estimation methods. We formulate this process as the parametric knowledge extraction:

$$\hat{\boldsymbol{\beta}} = \arg\min_{\boldsymbol{\beta}} \mathcal{L}_t \left( \boldsymbol{W}_0 + \sum_{k=1}^{t-1} \beta^k \boldsymbol{U}^k \boldsymbol{\Sigma}^k \boldsymbol{V}^k \right) \tag{5}$$

where $\boldsymbol{\beta} = [\beta^1, \ldots, \beta^{t-1}]^\top$ extracts optimal coefficients approximating the influence of new task data on each previous task SVD-based low-rank adapters. Then the parametric knowledge injection in the CL model initialization for task $\mathcal{T}_t$ is constructed as follows:

$$\boldsymbol{\theta}_t^{(0)} = \boldsymbol{W}_0 + \sum_{k=1}^{t-1} \boldsymbol{U}^k \boldsymbol{\Sigma}^k \boldsymbol{V}^k + \sum_{k=1}^{t-1} \hat{\alpha}^k \boldsymbol{U}^k \boldsymbol{\Sigma}^k \boldsymbol{V}^k, \text{ where } 1 + \hat{\alpha}^k = \hat{\beta}^k \tag{6}$$

In summary, while updating with low-rank models in CL can provide an implicit regularization effect to mitigate the forgetting issue, their generalization capability is limited due to the low-rank perturbation of parameters. This necessitates effective initialization to overcome the generalization limitations of low-rank updates. Specifically, an initialization that can effectively capture the shared knowledge across tasks based on their similarities to augment the knowledge captured by the pre-trained model is required to facilitate knowledge transfer and improve generalization when performing low-rank updates in a CL setting.

## 2.1 Learning from Knowledge Extraction and Injection

■ **Sequential Task Low-rank Adapter.** In our continual learning framework, for a pre-trained weight matrix $\boldsymbol{W}_0$, it parameterizes the incremental updates for task $\mathcal{T}_t$ by SVD-based low-rank matrices: $\boldsymbol{\theta}_t = \boldsymbol{\theta}_{t-1} + \boldsymbol{\Delta} = \boldsymbol{W}_0 + \sum_{k=1}^{t-1} \boldsymbol{U}^k \boldsymbol{\Sigma}^k \boldsymbol{V}^k + \boldsymbol{U}^t \boldsymbol{\Sigma}^t \boldsymbol{V}^t$. Similar to [50], we define $\mathcal{G}_{l,i}^t = \{\boldsymbol{U}_{l,*i}^t, \lambda_{l,i}^t, \boldsymbol{V}_{l,i*}^t\}$ as the $i$-th triplet ($i$-th singular value and its vectors) of layer $l$ for task $\mathcal{T}_t$, which is different from the dependent and not orthogonal doublet in LoRA-based low-rank matrices. With the SVD-based flexible regularizer, the training objective for task $\mathcal{T}_t$ is represented as $\mathcal{L}_t(\boldsymbol{U}^t, \boldsymbol{\Sigma}^t, \boldsymbol{V}^t) = \mathcal{L}_t(\boldsymbol{U}^t, \boldsymbol{\Sigma}^t, \boldsymbol{V}^t) + \gamma \sum_{i=1}^n \mathcal{R}(\boldsymbol{U}_i^t, \boldsymbol{V}_i^t)$, where $n$ is the number of weight matrices and $\gamma > 0$ is the regularization coefficient.

■ **Sensitivity of the Parameters.** Parameter sensitivity quantifies the impact on the loss function when a specific parameter is altered, typically set to zero, providing insight into the parameter's importance [14, 19, 26, 30, 53]. Consider a teacher model with parameters $\boldsymbol{\theta} = [\boldsymbol{\theta}_1, \dots, \boldsymbol{\theta}_{N_t}]$, where $N_t$ denotes the total number of parameters, the $i$-th parameter is $\boldsymbol{\theta}_i = [0, \dots, \boldsymbol{\theta}_i, \dots, 0]$. The sensitivity $S_{i,j}$ for sample $\boldsymbol{x}_j$ from task $\mathcal{T}$ is calculated as $S_{i,j} = |\boldsymbol{\theta}_i \nabla_{\boldsymbol{\theta}} \mathcal{L}(\boldsymbol{x}_j)| \approx |\mathcal{L}(\boldsymbol{\theta}) - \mathcal{L}(\boldsymbol{\theta} - \boldsymbol{\theta}_i)|$, where the approximation uses the first-order Taylor expansion of $\mathcal{L}(\cdot)$ at $\boldsymbol{\theta}_i$. Thus, we formulate the sensitivity of each triplet of low-rank matrices, rather than each parameter of the model. By masking the $l$-th layer $i$-th triplet $\mathcal{G}_{l,i}^k = \{\boldsymbol{U}_{l,*i}^k, \lambda_{l,i}^k, \boldsymbol{V}_{l,i*}^k\}$ of low-rank matrices for previous task $\mathcal{T}_k$, following [50], we can obtain the sensitivity score of the $l$-th layer $i$-th triplet of low-rank matrices for previous task $\mathcal{T}_k$, where $k \in \{1, \dots, t-1\}$, on new data sample $\boldsymbol{x}_t$ from new task $\mathcal{T}_t$:

$$S_{l,i}^k = S(\lambda_{l,i}^k) + \frac{1}{d_1} \sum_{j=1}^{d_1} S(\boldsymbol{U}_{l,ji}^k) + \frac{1}{d_2} \sum_{j=1}^{d_2} S(\boldsymbol{V}_{l,ij}^k) \tag{7}$$

■ **Knowledge Extraction and Injection.** For simplicity in analysis and mathematical expression, we assume that each layer within every task shares the same SVD-based low-rank adapter rank $r$. Our first step is to assess the layers of the previous task $\mathcal{T}_k$ low-rank matrices where $k \in \{1, \dots, t-1\}$, by Eq. 7, we calculate triplet-specific sensitivity scores in layer $l$ and this layer sensitivity score is represented as $S_l^k = \sum_{i=1}^r S_{l,i}^k$. To preserve the inherent structure of the tasks' low-rank adapters, the layers are subsequently mapped to the new task $\mathcal{T}_t$ adapter maintaining their original sequential order. Having computed the sensitivity scores of all triplets for layer $l$ in previous tasks $\{\mathcal{T}_1, \dots, \mathcal{T}_{t-1}\}$, we proceed to arrange them with weighted sensitivity scores to initialize the layer $l$ in new task $\mathcal{T}_t$:

$$\mathcal{G}_{l,i}^t = \{\boldsymbol{U}_{l,*i}^t, \lambda_{l,i}^t, \boldsymbol{V}_{l,i*}^t\} = \left\{ \sum_{k=1}^{t-1} \alpha_{l,i}^k \boldsymbol{U}_{l,*i}^k, \sum_{k=1}^{t-1} \alpha_{l,i}^k \lambda_{l,i}^k, \sum_{k=1}^{t-1} \alpha_{l,i}^k \boldsymbol{V}_{l,i*}^k \right\} \tag{8}$$

where $\alpha_{l,i}^k$ is the weight for the components of the triplet $\mathcal{G}_{l,i}^k$, and $\alpha_{l,i}^k = \frac{S_{l,i}^k}{\sum_{k=1}^{t-1} S_{l,i}^k}$ obtained from the sensitivity founded in Eq. 7. By aggregating weighted triplets across layers, we derive the extracted triplets from previous tasks' adapters and inject these triplets into the new task's adapter.

## 2.2 Training in Orthogonal Subspaces

We utilize the SVD-based adapter's low-rank subspace to represent the gradient subspaces of previous tasks, asserting that these parameters capture essential model update directions rather than just numerical adjustments [43]. This hypothesis allows us to minimize interference with previously learned tasks by training within a subspace orthogonal to that of the SVD-based subspace. We approximate the layer $l$ subspace for task $\mathcal{T}_k$ as the subspace consisting of the triplets $\boldsymbol{\mathcal{G}}_l^k$. To ensure orthogonal to the previously learned tasks layer subspaces, we first project the gradients of layer triplets of new task $\mathcal{T}_t$ onto the previously learned tasks layer subspaces spanned by $\{\boldsymbol{\mathcal{G}}_l^k\}$, where $k \in \{1, \dots, t-1\}$, and then make the gradients far away from these subspaces. The gradients of layer $l$ triplets of data $\boldsymbol{x}_t$ from new task $\mathcal{T}_t$ become:

$$\nabla_{\boldsymbol{\mathcal{G}}_l^t} \mathcal{L}_t(\boldsymbol{\mathcal{G}}_l^t; \boldsymbol{x}_t) = \nabla_{\boldsymbol{\mathcal{G}}_l^t} \mathcal{L}_t(\boldsymbol{\mathcal{G}}_l^t; \boldsymbol{x}_t) - \sum_{k=1}^{t-1} \mathsf{proj}(\nabla_{\boldsymbol{\mathcal{G}}_l^t} \mathcal{L}_t(\boldsymbol{\mathcal{G}}_l^t; \boldsymbol{x}_t), \boldsymbol{\mathcal{G}}_l^k) \tag{9}$$

where $\mathsf{proj}(\boldsymbol{u}, \boldsymbol{v}) = \frac{\langle \boldsymbol{u}, \boldsymbol{v} \rangle}{\langle \boldsymbol{v}, \boldsymbol{v} \rangle} v$ is the projection of $\boldsymbol{u}$ in the direction of $\boldsymbol{v}$ [7].

# 3 Experiments

## 3.1 Experimental Setup

Our experiments employ the encoder-decoder architecture of the T5-large and T5-base models [32], consistent with previous work in CL for NLP. All experiments are conducted on NVIDIA A6000 GPUs, utilizing the DeepSpeed repository.

■ **Standard CL benchmark.** We evaluate our approach using a CL benchmark specifically designed for language models. This benchmark comprises five text classification datasets: AG News, Amazon Reviews, Yelp Reviews, DBpedia, and Yahoo Answers, as introduced by [51]. We adhere to the CL setup for the T5 model as outlined in LFPT5 [31] and experiment with three different task orders within this benchmark.

■ **Large number of tasks.** Our method's efficacy is further tested on extended task sequences through a comprehensive CL benchmark encompassing 15 datasets, as detailed in [35]. This benchmark integrates tasks from three sources: five from the standard CL benchmark, four from the GLUE benchmark (MNLI, QQP, RTE, SST-2), five from the SuperGLUE benchmark (WiC, CB, COPA, MultiRC, BoolQ), and the IMDB movie reviews dataset. For each task, we train using 1000 randomly selected samples and validate using 500 samples per class, following the methodology of [35].

■ **Metrics.** We define the testing accuracy on the task $\mathcal{T}_i$ after training on the task $\mathcal{T}_j$ as $a_{i,j}$. The main metric for evaluation is **Average Accuracy (AA)**, calculated as the mean accuracy across all tasks after training on the last task: $\frac{1}{T} \sum_{i=1}^{T} a_{i,T}$.

■ **Baselines.** We compare our method against various baseline approaches:

- SeqFT [6]: train all model parameters on a sequence of tasks (without adding any regularization or replaying samples from the previous tasks).
- SeqLoRA: fixed-size LoRA parameters are trained on a sequence of tasks (without adding any regularization or replaying samples from the previous tasks).
- IncLoRA: incremental learning of new LoRA parameters on a sequence of tasks (without adding any regularization or replaying samples from the previous tasks).
- SeqSVD: fixed-size SVD parameters are trained on a sequence of tasks (without adding any regularization or replaying samples from the previous tasks).
- Replay: fine-tune the whole model with a memory buffer, and replay samples from old tasks when learning new tasks to avoid forgetting.
- EWC [13]: fine-tune the whole model with a regularization loss that prevents updating parameters that could interfere with previously learned tasks.
- LwF [18]: constrains the shared representation layer to be similar to its original state before learning the new task.
- L2P [44]: uses the input to dynamically select and update prompts from the prompt pool in an instance-wise fashion.
- LFPT5 [31]: continuously train a soft prompt that simultaneously learns to solve the tasks and generate training samples, which are subsequently used in experience replay.
- L-CL: incremental learning of new SVD parameters on a sequence of tasks with initialization and SVD regularization.
- B-CL: incremental learning of new SVD parameters on a sequence of tasks with gradient projection and SVD regularization.
- NLNB-CL: incremental learning of new SVD parameters on a sequence of tasks (with adding only SVD regularization and without replaying samples from the previous tasks), also called IncSVD.
- ProgPrompt [35]: adopts task-specific soft prompts for each task, training distinct models per task and using task IDs during inference.
- O-LoRA [43]: incrementally train new tasks in an orthogonal subspace while fixing the LoRA matrices of previous tasks.
- PerTaskFT: train a separate model for each task.
- MTL: train a model on all tasks as multi-task learning, serving as the benchmark's upper bound of the performance limit.

## 3.2 Main Results

Tab. 2 presents a performance comparison of LB-CL and baseline continual learning methods across two CL benchmarks. Following LFPT5, we report results from three random runs with different task orders on the CL benchmark. For fairness, we use the same rank of the LoRA-based and SVD-based matrix in each corresponding comparison experiment. To reduce computation time,

Table 2: Testing performance on two standard CL benchmarks with T5-large.

| | Standard CL Benchmark | | | | Large Number of Tasks | | | |
|---|---|---|---|---|---|---|---|---|
| | Order-1 | Order-2 | Order-3 | avg | Order-4 | Order-5 | Order-6 | avg |
| SeqFT | 18.9 | 24.9 | 41.7 | 28.5 | 7.4 | 7.3 | 7.4 | 7.4 |
| SeqLoRA | 39.5 | 31.9 | 46.6 | 39.3 | 4.9 | 3.5 | 4.2 | 4.2 |
| IncLoRA | 63.4 | 62.2 | 65.1 | 63.6 | 63.0 | 57.9 | 60.4 | 60.5 |
| SeqSVD | 40.0 | 63.3 | 44.9 | 49.4 | 13.7 | 13.8 | 12.2 | 13.2 |
| Replay | 50.3 | 52.0 | 56.6 | 53.0 | 54.5 | 54.3 | 53.5 | 54.1 |
| EWC | 46.3 | 45.3 | 52.1 | 47.9 | 44.9 | 44.0 | 45.4 | 44.8 |
| LwF | 52.7 | 52.9 | 48.4 | 51.3 | 49.7 | 42.8 | 46.9 | 46.5 |
| L2P | 59.0 | 60.5 | 59.9 | 59.8 | 57.7 | 53.6 | 56.6 | 56.0 |
| LFPT5 | 66.6 | 71.2 | 76.2 | 71.3 | 69.8 | 67.2 | 69.2 | 68.7 |
| L-CL | 75.3 | 73.5 | 71.9 | 73.6 | 66.5 | 64.0 | 69.0 | 66.5 |
| B-CL | 76.4 | 71.5 | 75.1 | 74.3 | 65.7 | 66.4 | 69.2 | 67.1 |
| NLNB-CL | 76.0 | 73.4 | 74.0 | 74.5 | 67.6 | 65.3 | 62.6 | 65.2 |
| O-LoRA | 74.9 | 75.3 | 75.9 | 75.4 | **70.5** | 65.5 | 70.5 | 68.8 |
| LB-CL | **76.9** | **76.5** | **76.8** | **76.7** | 68.4 | **67.3** | **71.8** | **69.2** |
| ProgPrompt | 76.1 | 76.0 | 76.3 | 76.1 | 78.7 | 78.8 | 77.8 | 78.4 |
| PerTaskFT | 70.0 | 70.0 | 70.0 | 70.0 | 78.1 | 78.1 | 78.1 | 78.1 |
| MTL | 80.0 | 80.0 | 80.0 | 80.0 | 76.3 | 76.3 | 76.3 | 76.3 |

we reasonably focus on some high-level layers to narrow the searching range of exploring critical parametric knowledge, since previous research, e.g. [43, 50], has already demonstrated that high-level layers are important for model performance.

■ **Results on Standard Continual Learning Benchmarks.** Across all task orders of the standard CL benchmark, LB-CL consistently surpasses previous methods by a significant margin. Notably, LB-CL achieves performance improvements in all task orders compared to O-LoRA, the prior state-of-the-art. Our approach demonstrates performance on par with multi-task learning and significantly exceeds that of PerTaskFT. This indicates that LB-CL not only effectively prevents catastrophic forgetting but also efficiently utilizes knowledge from prior tasks to enhance the learning of new tasks.

■ **Performance with Large Number of Tasks.** In a more demanding benchmark featuring a large number of tasks, LB-CL surpasses the state-of-the-art, O-LoRA, in terms of average performance across three task orders. Although ProgPrompt shows superior performance in managing long-sequence tasks, its limitations are notable. ProgPrompt is strictly dependent on the tasks it has been trained on and relies heavily on task IDs during inference, which restricts its generalization capabilities and adaptability for use in LLMs. However, LB-CL does not use task ID during testing, which keeps its generalization. It is important to note that nearly all existing continual learning methods fall significantly short of the performance achieved by PerTaskFT and MTL, underscoring that continual learning with a large array of tasks remains a formidable challenge.

### 3.3 Discussions

■ **How do initialization and gradient projection effectively influence the performance of LB-CL?**
We systematically evaluated the influence of two components: initialization and gradient projection. The results across various orders and their average performances, as presented in the last rows in Tab. 2, reveal distinct trends. L-CL, which employs only initialization, suggests that while this component does provide a beneficial starting point for task learning, it falls short of maintaining knowledge across tasks in Order 3. B-CL, utilizing only gradient projection, demonstrates slightly higher performance than L-CL in Order 1 and Order 3. However, the absence of initialization limits its effectiveness, particularly in establishing a robust foundation in Order 2. NLNB-CL, which neither initializes nor employs gradient projection, surprisingly performs slightly better than both L-CL and B-CL on average, but it does not excel in any specific order. This outcome suggests that the model may possess inherent adaptive capabilities or rely on other compensatory mechanisms. LB-CL, integrating both initialization and gradient projection, exhibits the highest overall performance with consistent scores across all orders. This robust performance indicates that the synergistic contribution of both components significantly enhances the model's ability to effectively handle CL tasks.

■ **How do different initialization strategies affect the performance?** To elucidate the impact of distinct initialization strategies, we compare two different initialization strategies motivated by [29] to initialize new task $\mathcal{T}_t$: (i) with previous tasks low-rank matrix triplets $\mathcal{G}^k$, and (ii) without the singular values $\mathbf{\Sigma}^k$ but leveraging $\{\mathbf{U}^k, \mathbf{V}^k\}$ where $k \in \{1, \ldots, t-1\}$. These strategies are assessed using T5-large model over three task orders in the standard CL benchmark. Our results, depicted in Fig. 3, reveal that utilizing only $\{\mathbf{U}, \mathbf{V}\}$ from prior tasks' triplets surpass the full triplet configuration in average performance across three task orders. While "with $\mathbf{\Sigma}$" strategy exhibits peak performance in Order 3, "without $\mathbf{\Sigma}$" approach demonstrates better consistency and stability. This suggests that excluding $\mathbf{\Sigma}$ may lead to a more robust generalization across diverse tasks, but it cannot fully represent previously learnt tasks' important subspaces, thus we use "with $\mathbf{\Sigma}$" as the initialization of LB-CL, and "without $\mathbf{\Sigma}$" can be used as an improvement strategy in implementation. Furthermore, both strategies outperform the performance of the O-LoRA method over these three orders.

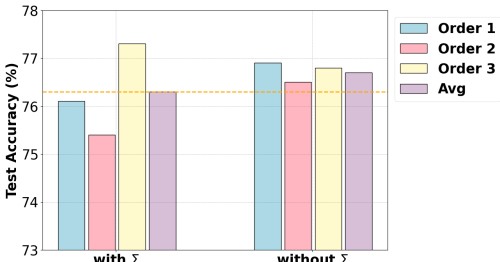

Figure 3: Comparison of different initialization strategies across three orders of standard CL benchmark. The "Avg" value represents the average testing accuracy, illustrating how each strategy stabilizes learning performance.

Figure 4: Impact analysis of seed sample quantity on the performance in LB-CL, evaluated across three orders of standard CL benchmark. This investigation highlights the influence of initial seed samples on model effectiveness.

■ **How does the number of seed samples affect the performance of the model?** The number of seed samples significantly influences the reliability and efficiency of the sensitivity score computations derived from the teacher model. We explored the impact of varying the number of seed samples from new tasks on the sensitivity of parameters from previous tasks. As illustrated in Fig. 4, increasing the number of seed samples improves performance incrementally at a slight rate. Notably, the variance in performance metrics is considerably lower with 4 and 8 seed samples. Based on these findings, we have selected 8 seed samples as the optimal number for our hyperparameter setting.

■ **How does the training computation cost perform?** We compare the training computation cost between LB-CL and O-LoRA in Tab. 3, using Order 1 in the standard CL benchmark with T5-large model. Tab. 3 shows that the GPU memory consumption of both methods is similar, indicating comparable resource efficiency during training. For the number of training parameters, we examine a single layer and denote $r$ as the rank of SVD-based and LoRA-based matrix, $m$ as the input dimension, and $n$ as the output dimension of the layer. Given that $r \ll \min(m, n)$, the number of training parameters for both methods remains close, further highlighting their efficiency in parameter.

Table 3: Comparison of training computation cost between LB-CL and O-LoRA.

| Method | GPU Memory | Num of training params/task |
|--------|------------|------------------------------|
| O-LoRA | 24.82 GB | $r(m+n)$ |
| LB-CL | 28.28 GB | $r(m+n)+r$ |

■ **What's the distribution of parametric knowledge across layers of the model?** We analyze the distribution of parametric knowledge across the model's layers to identify those most critical for retaining task-specific information. In Fig. 5, our analysis of sensitivity and Fisher information on the average of three task orders in standard CL benchmark, reveals that higher-level layers, particularly in decoder layers, exhibit significant sensitivity. Particularly, the top 4 layers of the decoder are notably sensitive, suggesting that focusing sensitivity analyses on these layers could represent the entire decoder effectively, thus reducing computational demands. Validation with Fisher information confirms that these high-level layers are crucial in both the encoder and decoder, especially the top 3 layers of the decoder. This alignment underscores that our sensitivity scores effectively identify the most crucial layers for task-specific knowledge transfer. Given the time-intensive nature of

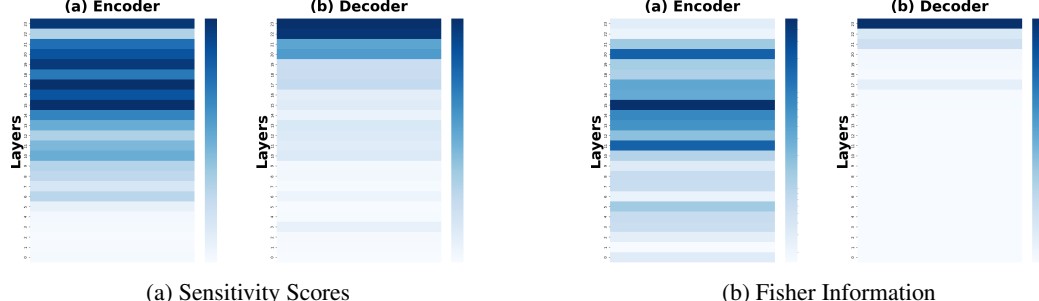

(a) Sensitivity Scores                    (b) Fisher Information

Figure 5: Comparison of sensitivity scores and Fisher information of encoder and decoder Layers, and both results are the average results of three task orders in standard CL benchmark.

Fisher information calculations, our sensitivity score approach provides a more efficient alternative, enhancing training efficiency by focusing on several critical high-level layers.

Table 4: Comparisons of different rank $r$ of low-rank matrix. This experiment is conducted based on T5-large in standard CL benchmark.

|  | **Order** | | | |
| **r-dim** | **1** | **2** | **3** | **avg** |
| 2 | 76.7 | 77.2 | 75.2 | 76.3 |
| 4 | 77.0 | 76.8 | 75.9 | 76.6 |
| 8 | 76.9 | 76.5 | 76.8 | 76.7 |
| 16 | 77.4 | 76.0 | 75.5 | 76.3 |
| **Std** | 0.25 | 0.44 | 0.60 | 0.18 |

Table 5: Comparisons of different models' performances across three task orders in standard CL benchmark.

| (T5-base) | **Order** | | | |
| **Method** | **1** | **2** | **3** | **avg** |
| O-LoRA | 72.9 | 72.3 | **72.6** | 72.6 |
| LB-CL | **73.8** | **74.4** | 72.4 | **73.5** |
| (T5-large) | **Order** | | | |
| **Method** | **1** | **2** | **3** | **avg** |
| O-LoRA | 74.9 | 75.3 | 75.9 | 75.4 |
| LB-CL | **76.9** | **76.5** | **76.8** | **76.7** |

■ **What's the optimal rank $r$ for LB-CL?** To explore the impact of the rank $r$ on the performance of LB-CL, we conduct experiments using the T5-large model on the standard CL benchmark. The results, presented in Tab. 4, examine how varying $r$ affects the accuracy across different task orders. It shows that increasing the rank $r$ does not lead to a significant improvement in model performance. Furthermore, the small standard deviations across different orders for each rank underscore the model's consistent performance, irrespective of rank variations. This suggests that by leveraging more knowledge from previous tasks, our method allows the gradient space of the new task to diverge more significantly from those of prior tasks, thereby enhancing stability across various ranks.

■ **How do different pre-trained models influence performance?** We investigate the impact of model scale on performance by comparing T5-base and T5-large models using a standard CL benchmark. We evaluate both our method and O-LoRA across three task orders. The results, presented in Tab. 5, clearly demonstrate significant performance differences between the two model sizes and the methods employed. For the T5-base model, LB-CL consistently outperforms O-LoRA. While for the T5-large model, LB-CL significantly surpasses O-LoRA's outcomes. Moreover, LB-CL shows exceptional consistency across all task orders in the T5-large model, highlighting its robustness and effectiveness when scaled up. This analysis confirms the influence of model size on the success of different continual learning strategies, with LB-CL proving particularly effective in larger models.

## 4  Conclusion

In this paper, we investigate the balance between overcoming forgetting and achieving generalization in the continual learning of LLMs, decompose the generalization error with the task low-rank matrix initialization, then propose a novel framework, exploring parametric knowledge transfer between tasks and utilizing the inherent forgetting less ability of low-rank matrix. Instead of storing extra task-specific auxiliary parameters, we just utilize the low-rank parameters which would be merged into the pre-trained model. Our experiments across standard CL benchmarks validate the effectiveness of this approach. Furthermore, we analyze the critical factors influencing initialization in CL, providing insights for further enhancements in this field.

## Acknowledgment

This work is partially supported by NSF CAREER Award #2239374 and NSF EFMA Award # 2318101.

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

# A    Appendix

## A.1    Additional Related Works

**Continual Learning.** Continual learning is a field focused on developing algorithms that can accumulate and refine knowledge over time, particularly as they encounter non-stationary data streams. The primary challenge in this domain is overcoming catastrophic forgetting, where the performance of a model on old tasks significantly diminishes when trained with new data. To address this, existing research has generally been divided into three main categories: (i) The *rehearsal-based methods* utilize a memory buffer to retain data samples from previous tasks, incorporating techniques such as experience replay [36], or constrained optimization to simultaneously train the model on current and previous tasks [8, 25]. (ii) The *regularization-based methods* introduce additional terms into the loss function to penalize changes in important model parameters, thereby limiting interference with previously learned tasks  [7, 13, 18, 38]. EWC [13] remembers old tasks by selectively slowing down learning on the weights important for those tasks. OGD  [7] constrains the parameters to move within the orthogonal space defined by the stored gradients of previous tasks. (iii) The *architecture-based methods* the aim is to minimize interference between tasks by dynamically expanding model capacity or creating isolated components for each task  [17, 33, 35, 37, 46]. Progressive Prompts [35] enhances forward transfer and mitigates catastrophic forgetting by learning a distinct prompt for each new task and sequentially appending these new task-specific soft prompts to the ones previously learned.

**Parameter-efficient Tuning.** Recent works on parameter efficient tuning (PET) [9] have shown that training only a subset of model parameters can yield performance comparable to training the full model, reducing computational demands and annotation efforts  [10, 11, 15, 48, 50]. BitFit [48] finds that updating only the bias-items during fine-tuning is very effective. Prompt tuning [15] uses learnable 'soft prompts' through back-propagation to condition frozen language models for specific tasks. LoRA [11] uses low-rank adapters in adapting models to new tasks with minimal additional parameters, and AdaLoRA [50] improves the performance of LoRA by adaptively allocating the parameter budget based on the weight matrices importance score. While the majority of PET has focused on learning one single task, there has been several efforts to apply PET to continual learning. AdapterCL [27] introduces an individual adapter block for each task. LFPT5 [31] utilizes a large soft prompt that is continuously trained on all tasks. O-LoRA [43] incrementally learns new tasks in orthogonal subspaces while keeping the LoRA parameters learned from previous tasks fixed to minimize catastrophic forgetting. InfLoRA [20] mitigates catastrophic forgetting by reparameterizing pre-trained weights with a small set of parameters, enabling fine-tuning within a subspace to maintain previous knowledge. The proposed MoE-Adapters and DDAS collaborate in [47] mitigates long-term forgetting by dynamically expanding a pre-trained CLIP model with Mixture-of-Experts adapters and preserves zero-shot recognition through a Distribution Discriminative Auto-Selector for routing in- and out-of-distribution inputs. ConPET [39] adapts existing continual learning strategies, originally developed for relatively smaller models to LLMs by incorporating PET with a dynamic replay approach. While O-LoRA addresses catastrophic forgetting through its incremental learning within orthogonal subspaces, it focuses on LoRA-based architecture rather than more general low-rank matrices and does not explore the knowledge transfer across different tasks.

## A.2    Implementation Details

All our experiments involving T5 models were performed on a server outfitted with four NVIDIA A6000 GPUs, utilizing the DeepSpeed repository for implementation. For every sequence of tasks across different orders, we standardized our experimental setup as follows: A constant rate of 1e-3 was maintained throughout the experiments. We used a total batch size of 32, distributed as 8 per GPU to leverage the computational capabilities of all four A6000 GPUs efficiently. We set the dropout rate at 0.1. We applied a regularization rate of 0.1 to the orthogonal matrices derived from the Singular Value Decomposition (SVD). A rate of 0.0 was employed, indicating no additional penalty on the model's weights during training.

## A.3    Datasets

Tab. 6 provides detailed information on the 15 datasets utilized in our continual learning (CL) experiments, including the evaluation metrics used for assessment. Our selection encompasses datasets

from established benchmarks:: the standard CL benchmark [51], GLUE [42], and SuperGLUE benchmarks [41], and added IMDB movie reviews dataset.

Table 6: The details of 15 datasets used in our CL experiments. NLI denotes natural language inference, QA denotes questions and answers task. The first five tasks correspond to the standard CL benchmark, all other tasks are used in long-sequence experiments.

| Dataset name | Category | Task | Domain | Metric |
|---|---|---|---|---|
| 1. Yelp | CL Benchmark | Sentiment Analysis | Yelp Reviews | Accuracy |
| 2. Amazon | CL Benchmark | Sentiment Analysis | Amazon Reviews | Accuracy |
| 3. DBpedia | CL Benchmark | Topic Classification | Wikipedia | Accuracy |
| 4. Yahoo | CL Benchmark | Topic Classification | Yahoo Q&A | Accuracy |
| 5. AG News | CL Benchmark | Topic Classification | News | Accuracy |
| 6. MNLI | GLUE | NLI | Various | Accuracy |
| 7. QQP | GLUE | Paragraph Detection | QUora | Accuracy |
| 8. RTE | GLUE | NLI | News, Wikipedia | Accuracy |
| 9. SST-2 | GLUE | Sentiment Analysis | Movie Reviews | Accuracy |
| 10. WiC | SuperGLUE | Word Sense Disambiguation | Lexical Databases | Accuracy |
| 11. CB | SuperGLUE | NLI | Various | Accuracy |
| 12. COPA | SuperGLUE | QA | Blogs,Encyclopedia | Accuracy |
| 13. BoolQA | SuperGLUE | Boolean QA | Wikipedia | Accuracy |
| 14. MultiRC | SuperGLUE | QA | Various | Accuracy |
| 15. IMDB | SuperGLUE | Sentiment Analysis | Movie Reviews | Accuracy |

Table 7: Six different task sequence orders utilized in continual learning experiments. Orders 1-3 follow the standard continual learning benchmark as established by previous research, focusing on a more traditional task sequence. Orders 4-6 customized for long-sequence experimentation, encompass 15 tasks each and are structured according to the methodologies outlined in [35].

| Order | Model | Task Sequence |
|---|---|---|
| 1 | T5-large,T5-base | dbpedia→ amazon → yahoo → ag |
| 2 | T5-large,T5-base | dbpedia→ amazon → ag→ yahoo |
| 3 | T5-large,T5-base | yahoo → amazon → ag → dbpedia |
| 4 | T5-large | mnli → cb → wic → copa → qqp → boolqa → rte → imdb → yelp → amazon → sst-2 → dbpedia → ag → multirc → yahoo |
| 5 | T5-large | multirc → boolqa → wic → mnli → cb → copa → qqp → rte → imdb → sst-2 → dbpedia → ag → yelp → amazon → yahoo |
| 6 | T5-large | yelp → amazon → mnli → cb → copa → qqp → rte → imdb→ sst-2 → dbpedia → ag → yahoo → multirc → boolqa → wic |

## A.4   Sensitivity Scores v.s. Fisher Information

In our analysis, we aim to elucidate the distribution of parametric knowledge intrinsic to different task orders and compare the sensitivity scores with Fisher Information. The insights from Fig. 6,7,8 indicate the distributions of parametric knowledge across layers remains consistent among different task orders. It suggests that regardless of the task sequence, the layer-wise distribution of parameters critical for task performance does not significantly vary. Both sensitivity scores and Fisher Information depict similar patterns, underscoring the robustness of our model's learning mechanisms. Our findings also highlight that higher-level decoder layers exhibit increased sensitivity compared to their lower-level counterparts. This heightened sensitivity in the decoder suggests that these layers play a more crucial role in refining the outputs, possibly due to their direct involvement in generating end task results. Notably, the sensitivity scores of the encoder layers in task order 3 are higher than those observed in the first two task orders. This variation could be attributed to the specific nature or complexity of the tasks in order 3, which might demand more nuanced feature extraction capabilities from the encoder layers.

Table 8: Instructions for different tasks

| Task | Prompts |
|---|---|
| NLI | What is the logical relationship between the "sentence 1" and the "sentence 2"? Choose one from the option. |
| QQP | Whether the "first sentence" and the "second sentence" have the same meaning? Choose one from the option. |
| SC | What is the sentiment of the following paragraph? Choose one from the option. |
| TC | What is the topic of the following paragraph? Choose one from the option. |
| BoolQA | According to the following passage, is the question true or false? Choose one from the option. |
| MultiRC | According to the following passage, is the question true or false? Choose one from the option. |
| WiC | Given a word and two sentences, whether the word is used with the same sense in both sentence? Choose one from the option. |

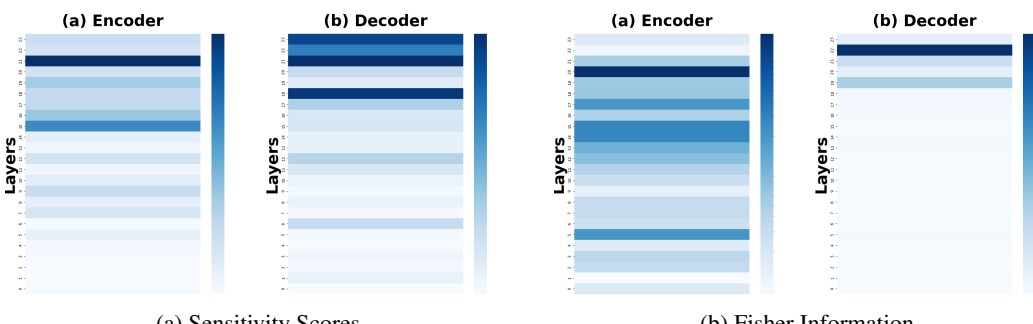

(a) Sensitivity Scores         (b) Fisher Information

Figure 6: Order 1: Sensitivity scores and Fisher information of encoder and decoder Layers

## A.5 Comparison of ROUGE score

We compare Average ROUGE-L scores (measures the longest common subsequence between the predicted and reference summaries, capturing sentence-level structure similarity) between O-LoRA and LB-CL on the standard CL benchmark in Tab. 9. It shows that the ROUGE-L scores of LB-CL achieve performance improvements across all three task orders of the Standard CL benchmark compared to O-LoRA, demonstrating the effectiveness of LB-CL.

Table 9: Comparison of ROUGE score between LB-CL and O-LoRA

| Method | Order 1 | Order 2 | Order 3 |
|---|---|---|---|
| O-LoRA | 0.7902 | 0.7868 | 0.7859 |
| LB-CL | 0.8169 | 0.8090 | 0.7894 |

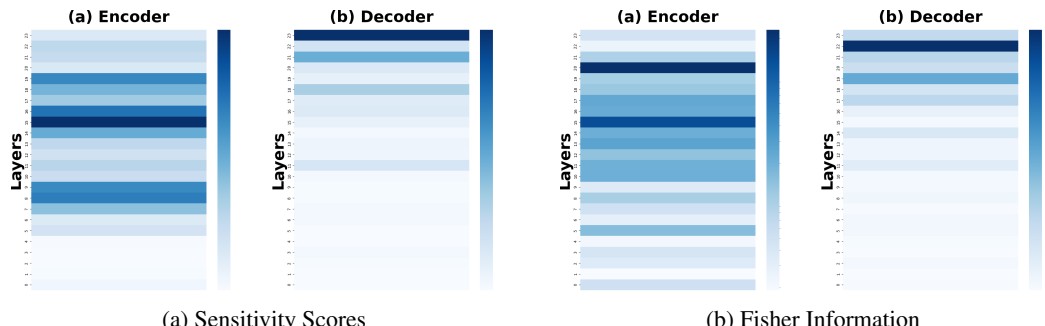

(a) Sensitivity Scores

(b) Fisher Information

Figure 7: Order 2: Sensitivity scores and Fisher information of encoder and decoder Layers

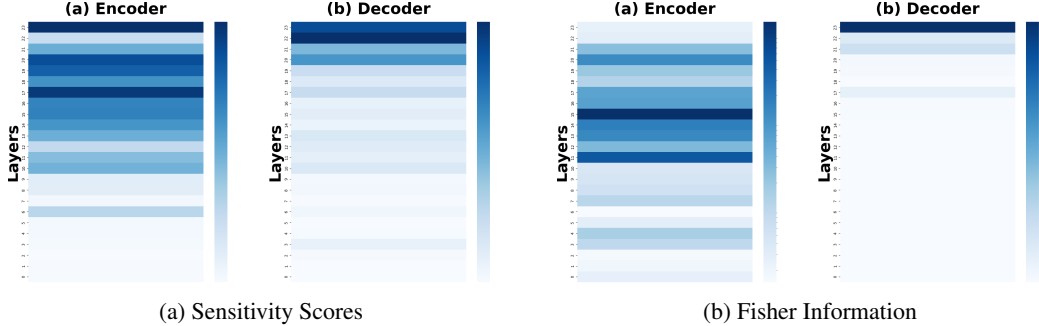

(a) Sensitivity Scores

(b) Fisher Information

Figure 8: Order 3: Sensitivity scores and Fisher information of encoder and decoder Layers

