# OpenReview forum: "Learn more, but bother less: parameter efficient continual learning"
_NeurIPS.cc/2024/Conference — NeurIPS 2024 poster_

### Official Review · Reviewer_qwD9 · 2024-07-04

**Soundness:** 3
**Presentation:** 3
**Contribution:** 2
**Rating:** 5
**Confidence:** 4

**Summary:**

The paper presents a new parameter-efficient method for continual learning in LLMs. The method focuses on two aspects of continual learning: (1) Catastrophic forgetting and (2) Forward transfer.  To address catastrophic forgetting the gradient update of the current task is performed on the orthogonal space of previous tasks. To address forward transfer, they propose to initialize the new SVD parameters of the current task using a combination of previous tasks via learned coefficients.  The method is tested on two benchmarks with three task orders. An ablation is provided for each proposed component. The method is compared to another parameter-efficient method for CL, variations of the proposed method, some basic baselines (MTL, standard finetuning), and few previous CL methods on standard architectures.

**Strengths:**

- Unlike most methods which focus on forgetting, the method addresses forward transfer as well.
- The method shows improvement in accuracy over a recent parameter-efficient method.
- Ablation and analysis of the proposed components and hyperparameters are provided.

**Weaknesses:**

- Despite that forgetting is one of the main aspects addressed by the paper, it is not evaluated and no forgetting metric is reported.
- Some very related works are missing [1,2].
- Some design choices/observations are not clear, see the questions section.
- Minor: Paper needs proofread. Suggestions for a few improvements are below:
    - L149: “since the the”
    - Section 3.1: A description of O-LoLA is missing in the baselines section
    - L 84 description of triplet could be provided here for clarity.
    - 3 rows (out of 4) from Table 3 are already presented in Table 2
    - Since the parameters of previous tasks are kept frozen, I am not sure whether forgetting is the right terminology or one should use interference (since forgetting is avoided by design somehow)

[1] Yu, Jiazuo, et al. "Boosting continual learning of vision-language models via mixture-of-experts adapters." Proceedings of the IEEE/CVF Conference on Computer Vision and Pattern Recognition. 2024.

[2] Liang, Yan-Shuo, and Wu-Jun Li. "InfLoRA: Interference-Free Low-Rank Adaptation for Continual Learning." Proceedings of the IEEE/CVF Conference on Computer Vision and Pattern Recognition. 2024.

**Questions:**

- Is the training performed in two stages? One for determining sensitivity parameters and one for training the task?
- Do you assume that the task identity is not available at the test time? If yes, is the input image at the test time forwarded to the sum of the SVD of all tasks?
- In Table 3, it is not clear to me why NLNB -CL is better than L-CL and B-CL, any thoughts on that?
- Do the same observations in Table 3 hold for the other benchmark?
- Figure 3 shows that excluding Sigma leads to better performance, I am wondering what could be the reasons behind this and why you don’t consider doing this in your proposed approach.
- How is your method compared with the mentioned works above?

**Limitations:**

Limitations and societal impact are not discussed.

---

> ### Author Rebuttal · Authors · 2024-08-07
>
> Thank you for your detailed feedback on our paper. We have thoughtfully addressed your insights and concerns, and hope our responses offer the necessary clarity on the issues raised.
>
> > **Q1: Training stage**
>
> Thanks for your interest. Yes, first we initialize the SVD parameters using sensitivity-determined important parameters for the new task (injecting knowledge), then fine-tuning the new task in orthogonal subspaces.
>
> > **Q2: Task identity not available at test time? If yes, is the input image at the test time forwarded to the sum of the SVD of all tasks?**
>
> Thank you for raising the concern. (1) Yes, task identity is not available at test time. (2) Inputs are forwarded through the whole model, including previously learned SVDs. We will make it clear in the subsequent revision.
>
> > **Q3: Performance of NLNB-CL, L-CL and B-CL**
>
> Thanks for your interest in this observation. As we discussed this result in section 3.3, NLNB-CL, while neither initializes nor employs gradient projection, performs slightly better on average testing accuracy but important to note that it does **NOT** achieve the best performance in any specific task order.
>
> - **Inherent adaptive capabilities**: pre-trained model with low-rank SVD matrices may have inherent adaptive capabilities or rely on other compensatory mechanisms, making performance more stable on average. This stability might lead to slightly better average testing accuracy compared to L-CL and B-CL.
> - **Orthogonal subspace in B-CL**: it preserves previous tasks' subspaces, retaining knowledge from previous tasks but somehow hindering optimization for new tasks, leading to suboptimal performance.
> - **Knowledge transfer in L-CL**: it emphasizes knowledge transfer, potentially disrupting previously learned subspaces, which may affect results.
>
> > **W4.4 \& Q4: Table 3**
>
> Thank you for raising the concern. No, Table 3 does not hold for the other benchmark. The reason we showed Table 3 separately is to compare each component (stage) clearly. We will remove Table 3 to avoid confusion due to repeated information in Table 2.
>
> > **Q5: About excluding Sigma**
>
> Thanks for your interest. Excluding Sigma and Including Sigma are different initialization strategies. Since the initialization of LB-CL idea is to learn previous tasks' important subspaces, constructed by triplets, if without $\boldsymbol{\Sigma}_i$, only doublets $\{\boldsymbol{U}_i,\boldsymbol{V}_i\}$ cannot fully represent these subspaces. We consider excluding Sigma as an initialization strategy, which can be used as an improvement strategy in implementation. We will clarify this in the revision.
>
> > **W2 \& Q6: Compare with the mentioned works above**
>
> Thank you for pointing out these related works. We appreciate the opportunity to compare our method with them.
>
> - **Model Focus**: [1] proposes a CL framework for a specific vision-language model, pre-trained CLIP, with Mixture-of-Experts (MoE) adapters. [2] proposes a LoRA-based continual learning method for a pre-trained Vision Transformer (ViT). However, our approach is for large language models and not a specific large language model.
>
> - **Method**:
>   - [1] uses MoE to dynamically activate the most related experts (adapters) by routers for a task, but the number of experts $N_{E}$ in the vision-language model is predefined and fixed, limiting flexibility and potentially affecting learning new tasks when none of the experts have the required knowledge. However, our approach uses a low-rank SVD matrix per task, learning tasks separately and mitigating forgetting.
>   - [2] algorithm not only stores previous LoRA parameters but also additionally stores gradient subspaces of previous tasks. Our approach initializes new task subspaces using previous SVD matrices without storing extra components, making it more memory-efficient. Also, large pre-trained models mainly fine-tune within a specific low-rank subspace, encapsulating crucial model update directions.
>
> - **Dataset**: [1] and [2] use image datasets, while our approach is applied to text (natural language) datasets.
>
> While [1] and [2] provide CL methods to vision-language models, our method offers a scalable and effective approach to CL in large language models, proven in NLP tasks. We are committed to including these related works and comparisons in the subsequent revision.
>
> > **W1: Evaluating forgetting and its metric**
>
> Thank you for mentioning this important aspect. We followed existing continual learning methods by evaluating model's performance primarily by average testing accuracy, which provides a good measure of how well the model performs on all tasks. In subsequent revisions, we will include additional metrics such as forward transfer score, backward transfer score, and forgetting metrics to offer a more thorough understanding of the model's behavior in CL.
>
> > **W4.1: Typo**
>
> Thanks for pointing out it. We will correct the typos in the subsequent revision.
>
> > **W4.2: Description of O-LoLA in baselines section**
>
> Thanks for pointing out this point. We will describe it in the subsequent revision, such as "O-LoRA: incrementally train new tasks in an orthogonal subspace while fixing the LoRA matrices of previous tasks."
>
> > **W4.3: Description of triplet**
>
> Thanks for mentioning this point. We will describe triplet in the subsequent revision in L 84, such as "a singular value and its corresponding vectors".
>
> > **W4.5: About catastrophic forgetting**
>
> Catastrophic forgetting occurs when a neural network learns new tasks, inadvertently overwriting or conflicting with knowledge of earlier tasks thus reducing performance. This is critical in continual learning, as it undermines the principle of consistently accumulating knowledge without negatively impacting prior learning. We compared NLNB-CL (only keeps SVD matrices of previous tasks frozen) with LB-CL. Results show performance degradation in NLNB-CL when learning new tasks, highlighting the need to address both forgetting and interference.

---

> > ### Comment · Reviewer_qwD9 · 2024-08-10
> > **Official comment by Reviewer qwD9**
> >
> > Thank you for your response. Some of my concerns have been addressed. I encourage the authors to consider the points raised by all reviewers in the revised version. I am leaning towards keeping my original score.

---

> > > ### Author Response · Authors · 2024-08-10
> > >
> > > Thank you for considering our responses. We are happy some of our answers were satisfying, and we promise that we will gladly incorporate the valuable and constructive comments raised by all reviewers in the revised version. We are also pleased to offer further clarification to make sure your remaining concerns are resolved.

---

### Official Review · Reviewer_aBm6 · 2024-07-12

**Soundness:** 2
**Presentation:** 3
**Contribution:** 2
**Rating:** 5
**Confidence:** 4

**Summary:**

This paper introduces LB-CL, a continual learning algorithm designed to tackle the issues of catastrophic forgetting and forward knowledge transfer. The approach integrates orthogonal low-rank SVD decomposition and sensitivity-based parameter initialization. The orthogonal subspace learning component addresses catastrophic forgetting by ensuring the SVD decompositions of different tasks remain orthogonal. The sensitivity-based initialization enhances forward transfer by optimizing parameter initialization weights. Comprehensive experiments across multiple representative datasets demonstrate that LB-CL outperforms many existing methods.

**Strengths:**

The paper introduces a novel approach that combines sensitivity-based knowledge transfer with orthogonal subspace learning, a unique method in the context of continual learning for LLMs.
The methodology is rigorously developed and supported by comprehensive experimental evaluations, demonstrating the effectiveness of LB-CL against existing state-of-the-art methods.
The experimental results robustly support the claims, showing that LB-CL outperforms state-of-the-art methods on standard continual learning benchmarks.

**Weaknesses:**

1. The paper does not sufficiently clarify the advantages and necessity of the proposed SVD decomposition over existing methods like LoRA.
2. The paper does not provide experiments to demonstrate the sensitivity of the method to the order of tasks.
3. The paper does not address whether the classification head is distinguished by task-ID during inference.
4. The paper's experimental section lacks a thorough comparison with other common initialization methods, which makes it difficult to assess the superiority of the proposed initialization strategy.
5. The computational complexity and scalability of maintaining orthogonal subspaces for a growing number of tasks are not addressed. As the number of tasks increases, the computational overhead may become prohibitive.

**Questions:**

Could you provide more details on why SVD decomposition is preferred over LoRA and what specific advantages it offers?
Have you tested the sensitivity of LB-CL to the order in which tasks are presented? How might task order impact the performance of the method?
How does the method perform when tasks are highly similar and distinguishing features overlap? Does the orthogonality constraint still effectively prevent forgetting in such cases?
During inference, does the model require the task-ID to be known? If so, how does this affect the practicality and flexibility of LB-CL?
The choice of rank in the low-rank SVD decomposition significantly impacts performance. How is this rank chosen optimally and how sensitive is the method to this hyperparameter?

**Limitations:**

*

---

> ### Author Rebuttal · Authors · 2024-08-07
>
> Thank you for your constructive feedback on our paper. We truly appreciate the time you invested in the review. We have carefully considered your insights and addressed the highlighted concerns. We hope our responses provide clarity on the matters raised.
>
>
> **W1 & Q1: About advantages and necessity of the proposed SVD decomposition preferred over LoRA**
>
> Thanks for your interest. SVD decomposition offers key advantages: (1) singular values are important for identifying the relationship between singular vectors in the orthogonal matrices, evaluating the importance of triplets ($\{\boldsymbol{U}_i,\boldsymbol{\Sigma}_i,\boldsymbol{V}_i\}$) efficiently, while $\boldsymbol{A}$ and $\boldsymbol{B}$ of LoRA are not orthogonal, making the doublets $(\{\boldsymbol{A}_i,\boldsymbol{B}_i\})$ dependent with each other and discarding the doublets can result in greater variation from the original matrix. (2) SVD matrix easily evaluates the importance of its components, making it more flexible, for example, SVD only masks singular values for pruning and maintains singular vectors, while LoRA may prune all elements if it is measured as unimportant, hindering reactivation. We will highlight this comparison in the subsequent revision.
>
> **W2 & Q2 & Q3: Sensitivity to task order**
>
> Thank you for your question. We have tested LB-CL's sensitivity to different task orders in original version of our paper. Table 2 (page 7) shows average testing accuracy for 3 different task orders in each CL benchmark. Let's revisit results on the standard CL benchmark for clarity:
>
> ***Task Order 1:*** dbpedia$\rightarrow$amazon$\rightarrow$yahoo$\rightarrow$agnews
>
> ***Task Order 2:*** dbpedia$\rightarrow$amazon$\rightarrow$agnews$\rightarrow$yahoo
>
> ***Task Order 3:*** yahoo$\rightarrow$amazon$\rightarrow$agnews$\rightarrow$dbpedia
>
> | **Method** | **Order 1** | **Order 2** | **Order 3** |
> |------------|-------------|-------------|-------------|
> | LB-CL      | 76.9%       | 76.5%       | 76.8%       |
>
>
> Results indicate a moderate sensitivity to task order. Achieving task order-invariance is challenging in continual learning, as the order of tasks can affect model performance. For example, learning Task A before Task B might yield different results compared to learning Task B before Task A.
>
> **Q4 & Q5: Perform when similar tasks and overlapping features, and whether prevent forgetting?**
>
> Thank you for raising this comment. (1) Our approach uses weighted triplets to transfer knowledge. When previous tasks are highly similar with distinguishing overlapping features for the new task, these tasks' important triplets for the new task are also similar. We use weighted scores for these triplets, constructing the new SVD matrix accordingly. (2) In such cases, the orthogonal gradient subspace is the intersection of these weighted triplets orthogonal subspaces, making the new task gradient orthogonal update to effectively prevent forgetting.
>
> **W3 & Q6 & Q7: During inference, does the model require task-ID to be known?**
>
> Thanks for mentioning this point. No, during the inference, we don't require task-ID. We use explicit instructions or demonstrations during training, injecting knowledge into the new SVD matrix, enabling the model to generalize well and handle unseen tasks efficiently. We are committed to emphasizing this in our subsequent revision.
>
> **Q8: Choice of rank in low-rank SVD decomposition and how sensitive is the method to rank?**
>
> Thanks for your interest. In Table 4 of original version of our paper (page 9), we compared average testing accuracy for different ranks of the low-rank SVD matrix. Let's revisit results for clarity:
>
>
> | **r-dim**  | **Order1**      | **Order2**     | **Order3**     | **Std.**  |
> |------------|------------|-----------|-----------|-----------|
> | 2          | 76.7       | 77.2      | 75.2      | 0.85      |
> | 4          | 77.0       | 76.8      | 75.9      | 0.48      |
> | 8          | 76.9       | 76.5      | 76.8      | 0.17      |
> | 16         | 77.4       | 76.0      | 75.5      | 0.80      |
> | **Std**    | 0.25       | 0.44      | 0.60      |           |
>
>
>
> It shows that in our scenarios, increasing rank does not significantly improve performance, and differences between ranks 2 and 16 are not significant. In our experiments, we used rank 8 for its smallest standard deviation, indicating consistent performance across different orders.
>
> **W4: Comparison with other common initialization methods**
>
> Thank you for your valuable feedback. We acknowledge the importance of thoroughly comparing our proposed initialization strategy with other common methods in continual learning to better assess its effectiveness. In our experiments, we compared our initialization strategy with a standard random initialization, using NLNB-CL as a baseline. This approach involves freezing SVD matrices of previous tasks and using a new randomly initialized SVD matrix for new tasks. This comparison highlighted our method's improvements. We are committed to providing a thorough evaluation of different initializations in subsequent revision, such as model-agnostic meta-learning.
>
> **W5: Computational complexity and scalability of maintaining orthogonal subspaces**
>
> Thanks for your insightful comment. As the number of low-rank SVD matrices grows, we merge updates into the initial parameters to mitigate GPU memory inflation, maintaining computational feasibility and preventing excessive memory usage. While our approach has shown effectiveness in empirical evaluations, its performance and scalability with a large number of tasks, such as hundreds, need further study. We will further focus on optimizing techniques and improving scalability to ensure practical use in extensive continual learning applications.

---

> > ### Comment · Reviewer_aBm6 · 2024-08-10
> > **Official Comment by Reviewer aBm6**
> >
> > Thank you for the positive response. Most of my doubts have been addressed. Therefore, I will keep my original rating.

---

> > > ### Author Response · Authors · 2024-08-10
> > >
> > > Thank you for checking our responses. We’re glad that most of your concerns have been addressed. We are also more than happy to make more clarifications that could address any remaining concerns and potentially increase the score.

---

### Official Review · Reviewer_X83x · 2024-07-14

**Soundness:** 4
**Presentation:** 3
**Contribution:** 3
**Rating:** 6
**Confidence:** 4

**Summary:**

This paper proposes the Learn More but Bother Less Continual Learning (LB-CL) algorithm for Continual Learning (CL) of Large Language Models. Unlike previous research, this paper introduces the idea of using SVD-based low-rank matrices to inject knowledge learned from previous tasks into new tasks, thereby enhancing plasticity during the CL process. Additionally, it suggests a method for learning in an orthogonal subspace to prevent forgetting of previous tasks. Experimental results on various text classification datasets demonstrate that the proposed algorithm outperforms existing algorithms. Furthermore, various ablation studies and analyses experimentally show the role and effectiveness of each component of the proposed algorithm.

**Strengths:**

The strengths of this paper are as follows:

1. The paper is well-written and easy to read overall. In particular, the explanation of the proposed algorithm in Section 2, "Generalization and Forgetting Tradeoff of Low-rank Finetuning," and Section 2.1 is seamlessly integrated, making it easy to understand the motivation and ideas behind the proposed algorithm in a natural flow.

2. The idea of Knowledge Extraction and Injection using SVD-based low-rank matrices adapters in CL to utilize knowledge from previous tasks for learning new tasks is novel and innovative.

3. The proposed algorithm demonstrated superior performance compared to existing algorithms in experiments conducted on various text classification benchmark datasets.

4. The extensive ablation studies and analyses experimentally validated the role and effectiveness of each component of the proposed algorithm, making this section particularly interesting to read.

**Weaknesses:**

LB-CL is motivated by O-LoRA and proposes a similar idea (Section 2.2: Training in Orthogonal Subspaces). Although the proposed algorithm achieves state-of-the-art performance in various experiments, the performance improvement over the previous SOTA algorithm, O-LoRA, is not very substantial (e.g., an average improvement of 1.3% on standard CL benchmarks and 0.4% on a large number of tasks). While LB-CL introduces additional ideas for knowledge transfer, this might increase the overall cost of the algorithm. Therefore, to verify the superiority of LB-CL over O-LoRA, it is necessary to compare the computation costs of these algorithms (e.g., training time, number of training parameters, or FLOPS).

Additionally, I have identified the following corrections that need to be made while reading the paper:

1. Line 192: It seems that Eq.5 should be corrected to Eq.7.
2. Line 224: It would be beneficial to include a brief explanation of O-LoRA under the 'Baselines' section.

**Questions:**

I find LB-CL proposed in this paper is somewhat novel. However, due to the similarity with the existing algorithm in certain aspects and the lack of significant performance improvement compared to the existing algorithm, I believe additional experiments and analysis are needed to properly validate the merits of the proposed idea. Please check the Weakness Section for more details.

**Limitations:**

There is no potential negative societal impact of this paper.

---

> ### Author Rebuttal · Authors · 2024-08-07
>
> Thank you for your appreciation and excellent summary of our work. We also appreciate the time and effort you dedicated to reviewing our research. We have addressed your questions and concerns below:
>
> > **W1: Line 192: It seems that Eq.5 should be corrected to Eq.7.**
>
> Thanks for your correction. Yes, Eq.5 should be corrected to Eq.7.
>
> > **W2: Line 224: It would be beneficial to include a brief explanation of O-LoRA under the ’Baselines’ section.**
>
> Thanks for your suggestions. We will include the explanation of O-LoRA should be briefly explained in the 'Baselines' section, such as "O-LoRA: incrementally train new tasks in an orthogonal subspace while fixing the LoRA matrices of previous tasks." in the subsequent revision.
>
> > **Q: Computation cost comparison**
>
> Thanks for your insightful suggestions.
>
> **Exp.Details**: We conduct computation cost comparisons on 4 NVIDIA A6000 GPUs and compare training costs between O-LoRA and LB-CL on task order 1 from the standard CL benchmark with T5-large model.
>
> | **Method** | **GPU Memory** | **Num of training params/task** |
> |------------|----------------|---------------------------------|
> | O-LoRA     | 24.82 GB       | $r(m+n)$                        |
> | LB-CL      | 28.28 GB       | $r(m+n)+r$                      |
>
> **Discussion:**
>
> - The GPU memory footprint of the two methods is quite close.
> - For the number of training params, we compare the trainable params within one layer. $r$ is the SVD matrix rank and LoRA rank, $m$ is the input dimension of the layer, and $n$ is the output dimension of the layer. Since $r \ll \min(m, n)$, the number of training params of two methods is also close.
> - LB-CL implementation computation cost is slightly more than O-LoRA. First, the orthogonal gradient update itself takes extra cost. Second, our code implementation computation cost is mainly incurred by DeepSpeed, a deep learning optimization library developed by Microsoft designed to train large-scale models efficiently. While we use DeepSpeed's `ZeRO stage 2' for efficient memory management, it currently does **NOT** support extracting gradients during training, as noted on their GitHub page. This limitation requires us to write an additional gradient computation module for orthogonal gradient update, increasing training costs. We plan to optimize our code implementation to reduce these costs.
>
> **Additional merits**:
>
> ROUGE (Recall-oriented Understudy for Gisting Evaluation) is a set of metrics designed to evaluate the quality of summaries by comparing them with reference summaries, which are widely used in natural language processing (NLP) tasks. ROUGE scores measure the overlap between the predicted and reference summaries, indicating how well the model-generated summaries capture the essential content of the reference summaries. We compare Average ROUGE-L scores (measures the longest common subsequence between the predicted and reference summaries, capturing sentence-level structure similarity) between O-LoRA and LB-CL on the standard CL benchmark:
>
> | **Method** | **Order 1** | **Order 2** | **Order 3** |
> |------------|-------------|-------------|-------------|
> | O-LoRA     | 0.7868      | 0.7759      | 0.7902      |
> | LB-CL      | 0.8169      | 0.7994      | 0.8090      |
>
>
> It shows that the ROUGH-L scores of LB-CL achieve performance improvements across all three task orders of the Standard CL benchmark compared to O-LoRA, demonstrating the effectiveness of LB-CL. Furthermore, the following table presents the average accuracy in Order 1 with different numbers of seed samples, which illustrates that LB-CL is more flexible and can achieve higher accuracy. We ultimately chose 8 seed samples in the original version of our paper because the variance across the 3 task orders was the least, as shown in Figure 4 of the original version of our paper (page 8).
>
> | **# Seed Sample** | **4**   | **8**   | **16**   | **64**   |
> |-------------------|---------|---------|----------|----------|
> | LB-CL             | 76.78%  | 76.90%  | 77.16%   | 77.32%   |
>
>
> We greatly appreciate your insightful feedback. Based on your valuable suggestions, we will include this computation cost analysis and provide a more detailed comparison of computation costs with other different algorithms. Additionally, we will incorporate the additional merits as discussed and include additional metrics such as the forward transfer score to offer a more thorough understanding of the model's behavior during continual learning in the experimental section of the subsequent version.

---

> > ### Comment · Reviewer_X83x · 2024-08-10
> >
> > Thank you for providing the author response. Almost all of my concerns have been addressed, so I keep my initial score 'weak accept'.

---

> > > ### Author Response · Authors · 2024-08-10
> > >
> > > Thank you for reviewing our responses. We appreciate your feedback and are glad that we could address almost all of your concerns. If you have any remaining questions or need further clarification, we are more than willing to provide additional clarification.

---

### Official Review · Reviewer_dHuW · 2024-07-23

**Soundness:** 3
**Presentation:** 3
**Contribution:** 3
**Rating:** 5
**Confidence:** 3

**Summary:**

This paper presents a novel approach to continual learning for large language models (LLMs). The proposed method, LB-CL, incorporates parameter-efficient tuning using low-rank subspace learning and orthogonal subspace projection to mitigate catastrophic forgetting. The study leverages incremental SVD-based low-rank matrix parameters for fine-tuning LLMs across a sequence of tasks. Comprehensive evaluations on benchmark datasets demonstrate the superiority of LB-CL over existing state-of-the-art methods in continual learning.

**Strengths:**

1. The use of incremental SVD-based low-rank matrix parameters and orthogonal subspace projection do address key challenges in continual learning such as catastrophic forgetting and efficient knowledge transfer.
2. The method is rigorously evaluated on multiple benchmark datasets, providing strong empirical evidence of its effectiveness. The baseline compared with this paper includes the latest papers published in 2024.
3. The paper includes an in-depth analysis of parametric knowledge transfer dynamics, initialization strategies, and the impact of seed samples on model performance, contributing to a deeper understanding of continual learning in LLMs.

**Weaknesses:**

1. The paper does not provide open access to the code and datasets used, which may hinder reproducibility and wider adoption of the proposed method.
2. The method in this paper is similar to O-LoRA [1] and has some improvements over O-Lora. This paper claims that "O-LoRA does not explicitly address knowledge transfer across different tasks." However, there is no specific evaluation in the experimental part of this paper to prove that the improvement of this paper on O-LoRA can improve "knowledge transfer across different tasks".
3. In the past, there have been some continual learning methods based on orthogonal subspaces in non-LLM fields, such as [2] and [3]. This paper does not point out the core differences between this method and these previous methods. Is this method just an application of past methods in the fields of LLM and LoRA?

[1] Wang, Xiao, et al. "Orthogonal subspace learning for language model continual learning." arXiv preprint arXiv:2310.14152 (2023).
[2] Chaudhry, Arslan, et al. "Continual learning in low-rank orthogonal subspaces." Advances in Neural Information Processing Systems 33 (2020): 9900-9911.
[3] Farajtabar, Mehrdad, et al. "Orthogonal gradient descent for continual learning." International Conference on Artificial Intelligence and Statistics. PMLR, 2020.

**Questions:**

see Weaknesses

**Limitations:**

see Weaknesses

---

> ### Author Rebuttal · Authors · 2024-08-07
>
> We sincerely appreciate your thorough review as well as constructive feedback. Your comments have been extremely helpful. We have carefully addressed each of your concerns and provided detailed answers to your questions below:
>
>
> > **W1: About open access to the code and datasets**
>
> We strongly agree with the importance of reproducibility and accessibility of our experiments and appreciate your suggestion. We have shared our code and datasets via an anonymized link with the Area Chair in a separate comment to maintain anonymity during the review process, per NeurIPS author response guidelines, which will be made public ultimately.
>
> > **W2: About specific evaluation on the improvement due to "knowledge transfer across different tasks"**
>
> Thank you for raising this comment. To specifically demonstrate the importance of knowledge transfer, we have conducted a detailed analysis of the influence of each component of our approach—initialization (knowledge transfer) and orthogonal gradient update—on overall performance in the original version of our paper. Table 3 presents the average testing accuracies under three scenarios:
>
> - **(1) No initialization (no knowledge transfer) and no orthogonal gradient update, called NLNB-CL**: This baseline scenario helps establish the effectiveness of our basic model without any enhancements for knowledge transfer or gradient management.
> - **(2) Only initialization (only knowledge transfer), called L-CL**: In this scenario, we focus on the impact of knowledge transfer. The results show an improvement in average testing accuracy, indicating that initializing new tasks with knowledge from previous tasks helps the model learn more effectively.
> - **(3) Only orthogonal gradient update, called B-CL**: This scenario isolates the effect of the orthogonal gradient update mechanism. The results demonstrate how managing gradient directions reduces interference between tasks, leading to better retention and performance.
>
> These three different scenarios demonstrate the importance of individual ingredients, which can show our approach improves the knowledge transfer across different tasks, compared to O-LoRA.
>
> > **W3: About core differences between this method and these previous continual learning methods based on orthogonal subspaces in non-LLM fields? Is this method just an application of past methods in the fields of LLM and LoRA?**
>
> Thank you for mentioning this point. We are happy to clarify the core differences between our method and previous methods and explain why our approach is not merely an application of past methods but a tailored solution for parameter-efficient continual learning in large language models (LLMs). We list the core differences in the following discussion:
>
> **Replay-based Approach in [2]**:
>
> - **Method**: [2] (Chaudhry et al.) use a memory buffer to store previous task data and replay it during the training of new tasks. It divides a random orthogonal space into several subspaces and allocates these subspaces one-to-one to each task with pre-defined orthogonal projections.
>
> - **Limitations**: Storing and replaying previous data can become impractical for large-scale models due to memory constraints and data privacy concerns.
>
> - **Our Approach**: We do not store any previous data, thereby ensuring data privacy and making our method more scalable for LLMs. Instead, we leverage the inherent low-rank structure of the model to manage orthogonal subspaces without relying on data replay.
>
> **Gradient Storage in [3]**:
>
> - **Method**: [3] (Farajtabar et al.) propose Orthogonal Gradient Descent (OGD), which stores a set of gradient directions in memory for previous tasks and projects new task gradients onto these stored orthogonal directions.
>
> - **Limitations**: Storing gradient directions for large-scale models is memory-intensive and impractical, especially as the number of tasks increases.
>
> - **Our Approach**: We avoid storing previous task gradients. Instead, we utilize low-rank subspaces to project new task gradients. This method is more memory-efficient and suitable for parameter-efficient continual learning in LLMs.
>
> In summary, both methods mentioned ([2] and [3]) involve storing either data or gradients, while our work addresses the unique challenges of parameter-efficient continual learning in LLMs by avoiding data and gradient storage, leveraging low-rank approximations, and ensuring data privacy. Our work is not just an application of past methods in the field of LLM and LoRA. We believe these core differences highlight the innovation and practicality of our approach. We will highlight these differences in the subsequent version.

---

> ### Comment · Area_Chair_oret · 2024-08-13
> **Please check authors' rebuttal**
>
> Dear reviewer,
>
> Please check authors' rebuttal they have made the effort to respond to your concerns.
>
> AC

---

### Author Rebuttal · Authors · 2024-08-07

Dear Reviewers,

We greatly appreciate your insightful feedback and valuable suggestions. We have provided specific responses to each reviewer’s questions separately. We sincerely thank you for your contributions to improving our work. If there are any further concerns or queries, we are fully prepared to address them.

Thank you for your time and effort.

---

### Decision · Program_Chairs · 2024-09-25

**Decision:**

Accept (poster)

**Comment:**

The paper proposes a method for continual learning of LLMs to defy forgetting based on LoRA. The approach is novel and considers the knowledge transfer and parameters relevance among different tasks which is an interesting direction for large overparameterized models.
The method shows SOTA performance on various continual learning benchmarks.
The concerns raised by reviewers were mostly addressed by the authors. We encourage the authors to include the clarification and the additional results and computation cost estimation in the final version.